# Genetic and Phenotypic Factors Affecting Glycemic Response to Metformin Therapy in Patients with Type 2 Diabetes Mellitus

**DOI:** 10.3390/genes13081310

**Published:** 2022-07-23

**Authors:** Yulia A. Nasykhova, Yury A. Barbitoff, Ziravard N. Tonyan, Maria M. Danilova, Ivan A. Nevzorov, Tatiana M. Komandresova, Anastasiia A. Mikhailova, Tatiana V. Vasilieva, Olga B. Glavnova, Maria I. Yarmolinskaya, Evgenia I. Sluchanko, Andrey S. Glotov

**Affiliations:** 1D. O. Ott Research Institute of Obstetrics, Gynaecology and Reproductology, 199034 Saint-Petersburg, Russia; yulnasa@gmail.com (Y.A.N.); barbitoff@bk.ru (Y.A.B.); ziravard@yandex.ru (Z.N.T.); elenamariamassa@gmail.com (M.M.D.); ban140598@gmail.com (I.A.N.); anamikhajlova@gmail.com (A.A.M.); o.glavnova@mail.ru (O.B.G.); m.yarmolinskaya@gmail.com (M.I.Y.); 2St. Petersburg State University, 199034 Saint-Petersburg, Russia; 3Pskov State University, 180000 Pskov, Russia; tatmyh005@mail.ru (T.M.K.); tatyana_korotche@mail.ru (T.V.V.); evgenija-sluchanko@rambler.ru (E.I.S.)

**Keywords:** type 2 diabetes mellitus, metformin, pharmacogenetics, therapy response variability, gene polymorphism, machine learning model

## Abstract

Metformin is an oral hypoglycemic agent widely used in clinical practice for treatment of patients with type 2 diabetes mellitus (T2DM). The wide interindividual variability of response to metformin therapy was shown, and recently the impact of several genetic variants was reported. To assess the independent and combined effect of the genetic polymorphism on glycemic response to metformin, we performed an association analysis of the variants in *ATM*, *SLC22A1*, *SLC47A1*, and *SLC2A2* genes with metformin response in 299 patients with T2DM. Likewise, the distribution of allele and genotype frequencies of the studied gene variants was analyzed in an extended group of patients with T2DM (*n* = 464) and a population group (*n* = 129). According to our results, one variant, rs12208357 in the *SLC22A1* gene, had a significant impact on response to metformin in T2DM patients. Carriers of *TT* genotype and *T* allele had a lower response to metformin compared to carriers of *CC*/*CT* genotypes and *C* allele (*p*-value = 0.0246, *p*-value = 0.0059, respectively). To identify the parameters that had the greatest importance for the prediction of the therapy response to metformin, we next built a set of machine learning models, based on the various combinations of genetic and phenotypic characteristics. The model based on a set of four parameters, including gender, rs12208357 genotype, familial T2DM background, and waist–hip ratio (WHR) showed the highest prediction accuracy for the response to metformin therapy in patients with T2DM (AUC = 0.62 in cross-validation). Further pharmacogenetic studies may aid in the discovery of the fundamental mechanisms of type 2 diabetes, the identification of new drug targets, and finally, it could advance the development of personalized treatment.

## 1. Introduction

Type 2 diabetes mellitus (T2DM) is a chronic metabolic disease characterized by insulin resistance and progressive pancreatic Beta cell dysfunction. The etiology of the disorder is known to have a significant genetic component that is confirmed by family- and twin-based studies [1]. Metformin is an oral hypoglycemic agent, a member of the biguanide class of drugs widely used as the first-line medication for the T2DM treatment, according to the clinical guidelines [2]. Despite the popularity of metformin in diabetes treatment, the exact mechanism of its action remains poorly understood and controversial [3]. The key target of metformin is thought to be the liver. The clinical studies in patients with T2DM confirmed the inhibition of hepatic glucose production (HGP) without concomitant increases in plasma insulin concentrations as the primary mechanism of action [4]. Metformin is taken up into the hepatocytes via the organic cation transporter 1 (OCT1). Metformin is supposed to inhibit the activity of the mitochondrial respiratory chain complex I, resulting in decreased ATP synthesis and an accumulation of AMP, which leads to the activation of AMP-activated kinase (AMPK) by promoting the formation of the AMPKαβγ heterotrimeric complex. Activation of the AMPK by metformin induces CBP phosphorylation at S436, resulting in the disassembly of the CREB-CBP-CRTC2 complex that causes the inhibition of gluconeogenic gene expression and HGP. In addition to the activation of classical AMPK signaling, several AMPK-independent mechanisms of action of metformin were also proposed [5,6,7]. The wide, interindividual variability of the responses to the therapy was recently shown for metformin treatment in various independent studies. Moreover, metformin therapy was associated with a higher rate of gastrointestinal symptoms (range 2–63% in different clinical trials) than most other oral antidiabetic agents. In approximately 4% of cases, this may cause the premature termination of therapy [8,9,10].

Over the past few years, several studies showed the association between the variants in the *ATM*, *SLC22A1*, *SLC47A1*, and *SLC2A2* genes and therapeutic responses to metformin. A protein encoded by the *ATM* gene belongs to the phosphatidylinositol 3-kinase family of proteins involved in the processes of DNA repair and/or cell cycle control. Mutations in this gene play a causative role in ataxia telangiectasia, an autosomal recessive disorder. The variant rs11212617 near the *ATM* gene was demonstrated to be associated with the response to metformin therapy in a GWA study in a GoDART (Genetics of Diabetes Audit and Research in Tayside) cohort of European ancestry, and in some other studies [11]. The *SLC22A1*, *SLC47A1*, and *SLC2A2* genes encode the transmembrane transporters OCT1, MATE1, and MATE2, respectively, which are known to be involved in the pharmacokinetics of a variety of environmental toxins and drugs, including metformin. Recent findings indicate that several polymorphic variants of these genes can make a significant contribution to the modulation of the glycemic response [12,13,14,15]. The discovery of the genetic determinants influencing the glycemic response can provide new knowledge of the underlying molecular mechanisms of T2DM patients’ response to therapy. That might finally allow progress in the therapy of T2DM and elaboration of the algorithms for tailored and precision treatment of this disorder. Despite the widespread incidence of T2DM in the Russian population [16], very few studies on the pharmacogenetics of T2DM were performed in Russia [17].

In this study, we aimed to analyze whether the genetic variants in the *ATM*, *SLC22A1*, *SLC47A1*, and *SLC2A2* genes influenced the glycemic response to metformin in a cohort of patients with T2DM in Russia and, furthermore, to reveal the integrated contribution of the genetic factors and phenotypic features to therapeutic response, using the machine learning approach.

## 2. Materials and Methods

### 2.1. Study Cohorts and Participants

A total of 464 unrelated patients with type 2 diabetes and 129 healthy volunteers were recruited. This study was performed using large-scale research facilities #3076082 “Human Reproductive Health”. Written informed consent for the research was obtained from all of the patients and healthy donors. T2DM was diagnosed based on the World Health Organization criteria. Patients with newly diagnosed diabetes mellitus (less than 1 year), type 1 diabetes, gestational diabetes, acute and/or decompensated liver and kidney disease, autoimmune disorders, malignancies, and under 30 years of age were excluded from the study. The level of HbA1c was determined in a fasting blood sample, body height and body weight were measured, and the body mass index (BMI) and waist–hip ratio (WHR) were calculated for all of the patients. The clinical parameters of the participants are shown in Table 1. Among the 464 T2DM patients examined, 299 patients were under continuous treatment with metformin for at least 6 months—131 individuals took it as monotherapy, and 168 took it in combination with other oral hypoglycemic agents. The metformin treatment response was estimated by an assessment of the decrease in HbA1c level after 6 months of therapy and by the achievement of the HbA1c individualized target.

### 2.2. DNA Isolation and Genotyping

The peripheral blood samples from patients were collected in tubes with EDTA. The genomic DNA was extracted from peripheral blood leukocytes, using the protocol for salt/chloroform DNA extraction with modifications [18]. The SNPs were selected based on the reported results of previous GWA and candidate gene studies. The information on the genetic variants analyzed in our study is shown in Table 2.

The SNPs were genotyped via polymerase chain reaction, followed by the restriction fragment length polymorphism (PCR-RFLP) method. Amplification was performed as follows: denaturation at 95 °C for 4 min; followed by 37 cycles of denaturation at 95 °C for 30 s; primer annealing at 60 °C for 30 s; and elongation at 72 °C for 30 s; and final extension at 72 °C for 5 min. The primers were designed by means of Oligo 6.0 software and NCBI BLAST tool. The sequences of the primers used and restriction enzymes for each analyzed SNP are given in Appendix A.

### 2.3. Statistical Analysis

#### 2.3.1. Association of Independent Variables with Response to Metformin Therapy

To determine whether the polymorphism of the *ATM*, *SLC22A1*, *SLC2A2*, and *SLC47A1* genes affects the therapy response, the genotype/allele frequencies were analyzed in four groups of metformin-treated patients: (i) metformin monotherapy and no response to treatment; (ii) metformin monotherapy/combination therapy and no response to treatment; (iii) metformin monotherapy/combination therapy and positive response to treatment; (iv) metformin monotherapy and positive response to treatment. To exclude possible biases in the results of the association analysis caused by ethnic differences, a comparative analysis of the frequencies of the genotypes/alleles of the *ATM*, *SLC22A1*, *SLC2A2*, and *SLC47A1* genes in the population cohort and the group of patients was carried out. Additionally, we compared the MAFs in the Russian population with reported data in European populations. For the comparison, we used the data presented in the resources 1000 Genomes Project and GnomAD (Genome and Exome sequence data) [28,29]. To analyze the association between the categorical variables, such as genotype/alleles frequencies and metformin response, as well as validating the Hardy–Weinberg equilibrium in the groups, Fisher’s exact test was used. A *p*-value ≤ 0.05 was considered to be statistically significant. All of the data were analyzed by using SPSS (SPSS Inc., Chicago, IL, USA) software. The odds ratios were estimated with 95% confidence intervals.

#### 2.3.2. Prediction of Response to Metformin Therapy Using Machine Learning

To predict the response to metformin therapy, we constructed a set of machine learning models, based on various sets of parameters. In total, 13 different variables were used in different combinations: fasting glucose; glycated hemoglobin (HbA1C), and creatinine levels in plasma; age and sex of the patient; BMI; WHR; familial T2DM background; and genotypes at five studied variant sites: rs11212617 in *ATM;* rs628031 and rs12208357 in *SLC22A1*; rs2289669 in *SLC47A1*; and rs8192675 in *SLC2A2*. Prior to model fitting, the SNP genotypes were transformed to numeric variables indicating the number of minor alleles at each locus. Next, Pearson’s correlation between the variables was assessed to test for predictor collinearity. No significant correlation between predictor variables was found, except for two LD-linked variants in *SLC22A1* (Appendix A).

The model training was performed using the caret package for R [30]. The patients with a positive response to metformin therapy were used as cases, and the non-responding patients were labeled as controls. Logistic regression was used as the main method for prediction; non-linear methods, such as decision tree and random forest, were also tried but showed a worse performance compared to regression. For the dimensionality reduction and feature selection, a Lasso regression approach [31] was used, with lambda values from 0.00068 to 0.125 used during the model tuning.

Due to the low sample size and a heavy class imbalance in the data, model training and validation were performed, using a custom permutation-based approach. At each step of the algorithm, a random set of 44 cases was drawn and then combined with 44 controls to construct a training set of 88 samples. Such a strategy allows us to train models with equal sizes of “positive’ and “negative” samples, which is known to increase the accuracy of model fitting. The obtained set of 88 samples at each step was used to train each type of model. For the model validation, two different metrics were then computed. First, the area under the receiver-operator curve (ROC/AUC) was evaluated, using a four-fold cross-validation (CV). While this value can be considered as the most unbiased measure of classifier performance, the splitting of the training data during CV might negatively affect the accuracy of the model fitting on smaller samples, decreasing the final estimate of model performance. Hence, we also used an alternative method of model evaluation. To do so, an independent non-overlapping set of 44 cases was drawn from the initial dataset and combined with the same 44 control individuals. The ROC/AUC value was then computed for such a ‘validation’ set (this approach is further referred to as “case-shuffling”).

Prior to further analysis, we evaluated the accuracy of the predictive model performance estimate by CV and case-shuffling methods. To this end, we attempted to classify random noise variables (different numbers of noise variables were simulated, ranging from 1 to 13). Expectedly, the mean AUC estimate in four-fold CV was close to 0.5 and did not depend on the number of predictors; at the same time, the AUC estimated using the training dataset was higher and increased substantially by adding random variables to the model (Appendix A). Most importantly, the mean estimated AUC in case-shuffling for classification using random noise was only slightly greater than for four-fold CV (mean AUC = 0.54) and did not depend on the number of predictor variables (Appendix A). As a result, the case-shuffling procedure can be used to compare the performance of different types of models, though the AUC = 0.54 value should be used as a corrected baseline.

To evaluate the importance of the predictor variables in the final models, scaled *t*-statistic values were computed, using the varImp function from the caret package. The data visualization was performed using the ggplot2 package [32].

## 3. Results

The distribution of the genotype frequencies in the group of T2DM patients and the control individuals followed the Hardy–Weinberg equilibrium. The allele frequencies of genetic variants of *ATM*, *SLC22A1*, *SLC47A1*, and *SLC2A2* genes in patients with T2DM and healthy controls are shown in Appendix A. The comparative analysis of the distribution of genotype/allele frequencies of the studied SNPs found no statistically significant differences between the patients with T2DM and the control group (*p*-value > 0.05). According to our results, the MAF of rs12208357 in the *SLC22A1* gene was found to be slightly statistically higher, and the rs2289669 in *SLC47A1* gene and rs8192675 in *SLC2A2* gene significantly lower than those in the European cohorts (*p*-value < 0.05). The MAFs of the rs628031 and rs2289669 variants in the Russian patient groups were commensurate with those of other European populations (*p*-value > 0.05) (Appendix A).

The distribution of the genotype and allele frequencies for genetic variants in the four groups of T2DM patients with different responses to metformin therapy is shown in Table 3. In all of the studied groups, the distribution of genotype frequencies followed the Hardy–Weinberg equilibrium. Comparison of the genotype and allele frequencies of the rs11212617 (*ATM*), rs628031 (*SLC22A1*), rs2289669 (*SLC47A1*), and rs8192675 (*SLC2A2*) variants in the responders and non-responders, we did not find any statistical differences. According to our results, the frequency of the *T* allele of the rs12208357 variant in the *SLC22A1* gene was increased in the group of non-responders in comparison with both the group of patients with a positive response to monotherapy/combination therapy with metformin (*p*-value = 0.0059), and the group with a positive response to monotherapy with metformin (*p*-value = 0.0418). Additionally, the association analysis of the rs12208357 variant with response to metformin showed that the *TT* genotype was statistically more common (*p*-value = 0.0250), and the *CC* genotype was significantly less common (*p*-value = 0.0246) in the non-responders, in comparison with the group with monotherapy/combination therapy.

To assess the influence of gene polymorphism on the glycemic response to metformin depending on the gender of the T2DM patients, we performed an analysis of the genotype and allele frequencies of studied variants in the groups of metformin responders and non-responders with different sex (male and female) (Appendix A). When comparing the genotype and allele frequencies of rs628031 (*SLC22A1*), rs2289669 (*SLC47A1*), and rs8192675 (*SLC2A2*) variants in male and female responders and non-responders, we did not find any significant differences. For the variant rs11212617 in *ATM* gene, we found a statistically significant increase in the frequency of the *CC* genotype and *C* allele in the female T2DM patients with glycemic response compared to the female non-responders (*p*-value = 0.0190 and *p*-value = 0.0386, respectively). Significant differences in the allele frequencies of the variant rs12208357 in *SLC22A1* gene were also found between the groups of male and female non-responders. The rare *T* allele was statistically more frequent in the male non-responders in comparison with the female non-responders (*p*-value = 0.0425).

We also asked whether the genotypes of the studied variants, as well as the additional phenotypic features, allowed for a prediction of the responses to metformin therapy in our patients. To answer this question, we built several predictive models based on: (i) the patient’s genotype at the rs12208357 variant in *SLC22A1* (the only variant that showed an association in the single variant tests); (ii) the genotypes at all of the tested variant sites; and (iii) the genotypes at all of the variant sites, as well as additional parameters, such as sex, age, BMI, WHR, plasma creatinine levels, and familial T2DM background. We also built a model that included fasting glucose and HbA1c levels as a positive control, as these variables should allow for a nearly complete discrimination between the cases and controls (Figure 1a; Appendix A). To evaluate the predictive performance of the models, we used a permutation-based approach, with two scoring metrics described in the Section 2.

All of the predictive models demonstrated a certain level of power to predict the response to metformin treatment (Appendix A). The genotype at the rs12208357 variant in the *SLC22A1* gene allowed the prediction of the response to metformin with a median AUC = 0.572 (all of the values are given with respect to the case-shuffling validation). A model that included all of the variant genotypes showed a slight performance gain over a single-variant model; at the same time, a model based on all of the genotypes and additional phenotypic features had a remarkably good performance (AUC = 0.772). As expected, a positive control model based on all of the 13 traits, including glycemic ones, showed nearly absolute predictive power (median AUC = 0.982).

We next went on to identify a set of parameters that had the greatest importance for the prediction of the response to metformin therapy. To do so, we decided to apply a regularization (Lasso) approach to construct a logistic regression model with the minimum necessary set of parameters. Similar to previous experiments, the Lasso fitting was performed in 1000 permutations to estimate the performance of the model and the importance of the model parameters. The Lasso approach successfully reduced the minimum necessary number of variables (average across 1000 permutation replicates = 2.4, median across the replicates = 2) while maintaining the general prediction accuracy. The analysis of the scaled parameter importance clearly identified a set of four features that had the highest impact on predictive power (Figure 1b): sex; rs12208357 genotype; familial T2DM background; and WHR. In concordance with these findings, a logistic regression model based on these four variables showed similar or better performance compared to the 11 parameter model (Figure 1a), and was the best model when model evaluation was performed using cross-validation (Appendix A, AUC = 0.62). While such a score does not make a high-accuracy model, it is notable that the four selected parameters allowed for a certain predictive power, even when using a more rigorous validation approach.

## 4. Discussion

The glucose-lowering effect of metformin was previously shown to have a wide interindividual variability, due to numerous causes including genetic factors [33]. The recent progress in the identification of variants in the *ATM*, *SLC22A1*, *SLC2A2*, and *SLC47A1* genes associated with the therapeutic response to metformin demonstrated their potential involvement in the metformin action mechanisms. However, these findings were not replicated in some cohorts, while a number of studies even received controversial results [11,12,19,20,26,27,34]. This inconsistency of the results can be explained by the differences in the study design, including the size and characteristics of the cohort, type of study, methods of analysis, and methods of estimation of the treatment effectiveness, as previously discussed [35].

Firstly, to explore whether the variants rs11212617, rs628031, rs12208357, rs2289669, and rs8192675 could be involved in the genetic susceptibility to T2DM in the Russian population, we performed a comparative analysis for the allele and genotype frequencies for all of the studied variants between the group of patients with T2DM and the sample of healthy controls in the cohort with Russian ethnicity. The results of this study found no association of the variants with T2DM development in Russia. Secondly, we analyzed the independent association of each of the studied variants with the metformin response for the patients categorized into four groups: (i) all responders vs. non-responders; (ii) metformin-responding patients (taking monotherapy only) vs. non-responders; (iii) all responders vs. non-responders (taking monotherapy only); and (iv) metformin-responding patients (taking monotherapy only) vs. non-responders (taking monotherapy only). According to the results of our study, no association was found between the variants in the *ATM* (rs11212617), *SLC22A1* (rs628031), *SLC47A1* (rs2289669), and *SLC2A2* (rs8192675) genes and the therapeutic response to metformin in the Russian patients with T2DM. The same results were obtained in the previous studies: for the rs11212617 variant in a set of studies in different European and Asian populations [36,37,38]; for rs628031 in European and Iranian populations [34,39]; for rs2289669 in European [40,41] and Indian populations [42,43]; for rs8192675–in the cohort of Action to Control Cardiovascular Risk in Diabetes (ACCORD) [38]. Based on the data that the association of these variants with the metformin response were not confirmed in some ethnic cohorts, the certain genetic heterogeneity for the polymorphic variants in the *ATM* (rs11212617), *SLC22A1* (rs628031), *SLC47A1* (rs2289669), and *SLC2A2* (rs8192675) genes between the different populations could be assumed. Our results showed the statistically significant differences in the MAFs of rs12208357 (*SLC22A1*), rs2289669 (*SLC47A1*), and rs8192675 (*SLC2A2*) between the Russian population and the European cohorts, which also is supposed to be an argument in favor of the point about the possible interethnic differences in the distribution of these gene alleles.

One of the studied genes, *SLC22A1*, is known to encode for a polyspecific organic cation transporter 1 (OCT1), a member of the solute carrier 22 family of transporter proteins which are involved in the absorption, distribution, metabolism, and excretion of several organic cations. OCT1 is predominantly expressed in the liver, but it is also found in other tissues [44,45]. It was shown to be essential for the uptake of metformin by the hepatocytes [46]. OCT1 was figured out to be highly polymorphic in ethnically diverse populations. As shown in our study, the carriers of *TT* genotype and *T* allele of rs12208357 (*SLC22A1* gene) had a lower response to metformin therapy compared to the carriers of the *CC/CT* genotypes and *C* allele (*p*-value = 0.0246, *p*-value = 0.0059, respectively). Our data are consistent with the previously obtained results which determine this variant as affecting the treatment response, including that through the hepatic exposure of metformin [12,47]. The genetic variability of the *SLC22A1* gene was previously shown to possibly influence the functioning of the OCT1 protein and, therefore, to modulate the pharmacokinetics and therapeutic response to metformin. The rs12208357 in the *SLC22A1* gene is a nonsynonymous missense variant located in exon 1 of the gene, which results in the loss of the OCT1 protein function and causes the reduced uptake of metformin and 1-methyl-4-phenylpyridinium (MPP(+)) [12,48,49]. Thereby, according to our results, the variant rs12208357 in the *SLC22A1* gene might independently impact the glucose-lowering effect of metformin in the Russian population.

In the present study, we next questioned whether the combinations of the genotyped variants and other affecting factors (age, gender, BMI, WHR, creatinine level) were associated with the metformin response, and whether the patient’s genotype could be used to predict the response to the metformin therapy. To answer these questions, we applied a machine learning-based approach and constructed a series of classification models to predict the response to metformin treatment, either alone (monotherapy vs. non-responding patients) or combined with other therapeutics (all metformin-responding patients vs. non-responders). A model based on the data of all of the studied gene variants and phenotypic parameters had the best performance, whereas the predictive power of the model including only SNPs did not differ from the model based on a single variant rs12208357. This may be caused by both the possibly large contribution of environmental factors or the small influence of each specific genetic variant in the metformin response; additionally, the population characteristics of the Russian cohort and insufficient sample size can be the logical explanation for these results. Finally, we were able to construct an optimal model demonstrating that a set of four parameters taken together (gender, rs12208357 genotype, familial T2DM background, and WHR) had the greatest importance for the prediction of the response to metformin therapy in Russian patients. According to the data obtained, male gender, *TT* genotype of the variant rs12208357 (*SLC22A1*), familial T2DM background, and increased WHR were found to be associated with the reduced metformin response in our population.

Interestingly, the study also showed a gender-specific difference in response to metformin treatment for the carriers of the *CC* genotype of rs11212617 (*ATM*) and *TT* of rs12208357 (*SLC22A1*). Some evidence was found for the potential effect of metformin on blood glucose levels in a sex-dependent manner. Li and colleagues suggested a better glycemic response during treatment in females compared with that of males in Chinese patients. Moreover, an increase in the insulin secretion was determined in the same study in the female patients, whereas the males displayed no significant change [50]. However, the results of a large German study demonstrated that metformin treatment had different effects on body weight and HbA1c between females and males. Women showed a significantly higher reductions in body weight after treatment, whereas men displayed significantly higher HbA1c-reductions after metformin monotherapy treatment [51]. In a Chinese cross-sectional study that included metformin-treated patients with T2DM, the association between the rs622342 variant and HOMA-IR and the association between rs11212617 variant and HOMA-BCF were gender-dependent [52]. Thus, gender may represent an important factor for the process of determining the individualized treatment goals and the assessment of therapy results, but large-scale studies based on the estimation of many genetic and phenotypic parameters are needed to understand gender differences in action and the pharmacokinetics of metformin.

Thereby, our results demonstrate the utility of genotypic and phenotypic information for predicting the response to metformin therapy, which could undoubtedly contribute to the improvement of the treatment strategy of T2DM patients. Further studies of the aforementioned factors and the other genotypic and phenotypic parameters in different, large and ethnically homogeneous cohorts may be useful to confirm the findings and to improve the predictive power of the model constructed.

## Figures and Tables

**Figure 1 genes-13-01310-f001:**
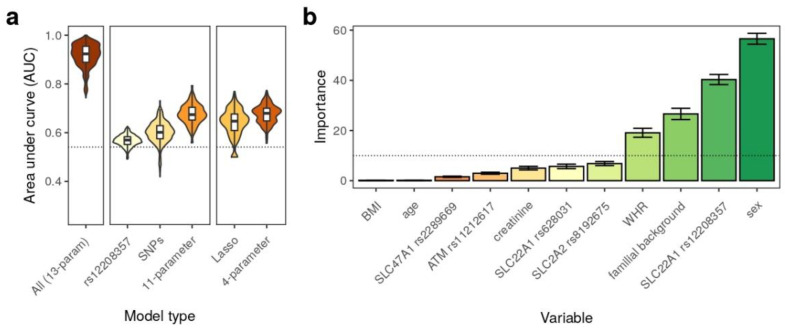
Prediction of response to metformin treatment using genotypes and additional phenotypic features. (**a**) Area under receiver-operator (ROC) curve (AUC) values for different types of models based on all parameters (including glycemic traits), rs12208357 genotype alone, all SNP genotypes (SNPs), 11-parameter model (all features excluding glycemic traits), as well as the Lasso regression model and a final 4-parameter model based on sex, rs12208357, WHR, and familial T2DM background. The dashed line indicates an AUC = 0.54 threshold, corresponding to the performance of random noise classification in case-shuffle test (see Materials and Methods section for details); (**b**) Scaled importance of features in the Lasso regression model. Mean estimates across 1000 replicates are shown. Error bars correspond to the *t*-based 95% confidence interval for the mean. The dashed line represents a 10% scaled importance threshold.

**Table 1 genes-13-01310-t001:** Clinical characteristics of T2DM patients and healthy controls.

Characteristics	All T2DM Patients (*n* = 464)	T2DM Patients Taking Metformin (*n* = 299)	Controls (*n* = 129)
**Male (*n*)**	155	93	91
**Female (*n*)**	309	206	38
**Age (years)**	61.11 ± 13.65	60.92 ± 13.00	40 ± 14.28
**BMI (kg/m^2^)**	31.96 ± 8.2	32.58 ± 6.51	24.43 ± 2.79
**FBG (mmol/L)**	7.88 ± 2.4	7.87 ± 2.36	4.66 ± 0.37
**Family history of diabetes (*n*)**	158	108	0
**Creatinine (mmol/L)**	0.09 ± 0.03	0.09 ± 0.02	NA
**WHR**	0.98 ± 0.097	0.95 ± 0.090	NA
**HbA1c (%)**	7.53 ± 1.14	7.44 ± 1.17	NA

Data are expressed as mean ± SD; BMI: body mass index; FBG: fasting blood glucose; WHR: waist–hip ratio; NA: not assessed.

**Table 2 genes-13-01310-t002:** Genetic variants analyzed in the study.

#	Gene Symbol	Region	dbSNP ID	Nucleotide Change	Amino Acid Change	Function	References
1	*ATM*	11q22.3	rs11212617	intron C/A	-	↑	[11,19,20,21]
2	*SLC22A1*	6q25.3	rs628031	c.1222A > G	Met408Val	↑ SE	[22,23]
3	*SLC22A1*	6q25.3	rs12208357	c.181C > T	Arg61Cys	↓	[12]
4	*SLC47A1*	17p11.2	rs2289669	intron G/A	-	↑↓	[24,25]
5	*SLC2A2*	3q26.2	rs8192675	intron A/G	-	↑	[26,27]

↑—increased response to therapy (in relation to the minor allele); ↓—reduced response to therapy (in relation to the minor allele); SE—side effects.

**Table 3 genes-13-01310-t003:** Genotype and allele frequencies of studied variants in patients with glycemic response to metformin treatment and non-responder patients.

Genotype/ Allele	Patients with Glycemic Response		Non-Responder Patients		*p*-Value
	(Monotherapy/Combination Therapy) *n* (%)	(Monotherapy) *n* (%)	(Monotherapy/Combination Therapy) *n* (%)	(Monotherapy) *n* (%)	
**rs11212617 *ATM***					
** *AA* **	66 (26)	37 (31)	11 (25)	5 (42)	*p**_1_* = 1.0000 *p**_2_* = 0.5622 *p**_3_* = 0.3126 *p**_4_* = 0.5208
** *AC* **	133 (52)	61 (51)	25 (57)	4 (33)	*p**_1_* = 0.6255 *p**_2_* = 0.5975 *p**_3_* = 0.2455 *p**_4_* = 0.3646
** *CC* **	56 (22)	21 (18)	8 (18)	3 (25)	*p**_1_* = 0.6923 *p**_2_* = 1.0000 *p**_3_* = 0.7310 *p**_4_* = 0.4599
** *A* **	265 (52)	135 (57)	47 (53)	14 (58)	*p**_1_* = 0.8182 *p**_2_* = 0.6168 *p**_3_* = 0.6768 *p**_4_* = 1.0000
** *C* **	245 (48)	103 (43)	41 (47)	10 (42)	
**rs628031 *SLC22A1***					
** *AA* **	28 (11)	13 (11)	8 (18)	0 (0)	*p**_1_* = 0.2080 *p**_2_* = 0.2908 *p**_3_* = 0.6219 *p**_4_* = 0.6079
** *AG* **	128 (50)	64 (54)	22 (50)	7 (58)	*p**_1_* = 0.1000 *p**_2_* = 07253 *p**_3_* = 0.7695 *p**_4_* = 1.0000
** *GG* **	98 (39)	42 (35)	14 (32)	5 (42)	*p**_1_* = 0.5004 *p**_2_* = 0.7143 *p**_3_* = 1.0000 *p**_4_* = 0.7549
** *A* **	184 (36)	90 (38)	38 (43)	7 (29)	*p**_1_* = 0.2329 *p**_2_* = 0.4435 *p**_3_* = 0.5236 *p**_4_* = 0.5081
** *G* **	324 (64)	148 (62)	50 (57)	17 (71)	
**rs12208357 *SLC22A1***					
** *CC* **	219 (87)	100 (84)	32 (73)	11 (92)	***p_1_* = 0.0250** *p**_2_* = 0.1179 *p**_3_* = 1.0000 *p**_4_* = 0.6913
** *CT* **	32 (13)	18 (15)	9 (20)	1 (8)	*p**_1_* = 0.1627 *p**_2_* = 0.4776 *p**_3_* = 1.0000 *p**_4_* = 1.0000
** *TT* **	2 (1)	1 (1)	3 (7)	0 (0)	***p******_1_* = 0.0246** *p**_2_* = 0.0604 *p**_3_* = 1.0000 *p**_4_* = 1.0000
** *C* **	470 (93)	218 (92)	73 (83)	23 (96)	***p**_1_* = 0.0059** ***p**_2_* = 0.0418** *p**_3_* = 1.0000 *p**_4_* = 0.7036
** *T* **	36 (7)	20 (8)	15 (17)	1 (4)	
**rs2289669 *SLC47A1***					
** *AA* **	37 (14.5)	20 (17)	11 (25)	4 (33)	*p**_1_* = 0.1163 *p**_2_* = 0.2638 *p**_3_* = 0.0940 *p**_4_* = 0.2312
** *AG* **	88 (34.5)	36 (30)	10 (23)	1 (8)	*p**_1_* = 0.1637 *p**_2_* = 0.4340 *p**_3_* = 0.0667 *p**_4_* = 0.1773
** *GG* **	130 (51)	63 (53)	23 (52)	7 (59)	*p**_1_* = 1.0000 *p**_2_* = 0.8623 *p**_3_* = 0.7702 *p**_4_* = 0.7705
** *A* **	162 (32)	76 (32)	32 (36)	9 (37.5)	*p**_1_* = 0.3911 *p**_2_* = 0.5078 *p**_3_* = 0.6547 *p**_4_* = 0.6485
** *G* **	348 (68)	162 (68)	56 (64)	15 (62.5)	
**rs8192675 *SLC2A2***					
** *AA* **	147 (58)	71 (60)	20 (45)	5 (42)	*p**_1_* = 0.1404 *p**_2_* = 0.1134 *p**_3_* = 0.3720 *p**_4_* = 0.3578
** *AG* **	90 (35)	39 (33)	21 (48)	7 (58)	*p**_1_* = 0.1306 *p**_2_* = 0.0998 *p**_3_* = 0.1293 *p**_4_* = 0.1110
** *GG* **	17 (7)	9 (7)	3 (7)	0 (0)	*p**_1_* = 1.0000 *p**_2_* = 1.0000 *p**_3_* = 1.0000 *p**_4_* = 1.0000
** *A* **	384 (76)	181 (76)	61 (69)	17 (71)	*p**_1_* = 0.2323 *p**_2_* = 0.2538 *p**_3_* = 0.6287 *p**_4_* = 0.6190
** *G* **	124 (24)	57 (24)	27 (31)	7 (29)	

*p*-value ≤ 0.05 was considered statistically significant and is shown in bold; *p_1_*: *p*-value calculated for all patients with glycemic response (taking metformin as monotherapy/as combination therapy) compared to all non-responder patients (taking metformin as monotherapy/as combination therapy); *p_2_*: *p*-value calculated for patients with glycemic response (taking metformin as monotherapy only) compared to all non-responder patients (taking metformin as monotherapy/as combination therapy); *p_3_*: *p*-value calculated for all patients with glycemic response (taking metformin as monotherapy/as combination therapy) compared to non-responder patients (taking metformin as monotherapy); *p_4_*: *p*-value calculated for patients with glycemic response (taking metformin as monotherapy) compared to non-responder patients (taking metformin as monotherapy).

## Data Availability

Publicly available datasets were analyzed in this study. This data can be found here: https://github.com/mrbarbitoff/metformin-response, accessed on 20 July 2022.

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
