# Peer review of "Genetic and Phenotypic Factors Affecting Glycemic Response to Metformin Therapy in Patients with Type 2 Diabetes Mellitus"

_genes, 2022, doi:10.3390/genes13081310_

Round 1

Reviewer 1 Report

The manuscript of “Genetic and phenotypic factors affecting glycemic response to metformin therapy in patients with type 2 diabetes mellitus” by Y.A. Nasykhova and co-authors aims to study the association between genetic variants in ATM, SLC22A1, SLC47A1, and SLC2A2 and therapeutic response to metformin, as well as the distribution of allele and genotype frequencies of these gene variants in a population cohort. The gene-based analysis indicated that one variant, rs12208357 in SLC22A1 gene (that encodes for a polyspecific organic cation transporter 1 (OCT1), had an impact on response to metformin in T2DM patients, with the carriers of TT genotype and T allele of rs12208357 had a low response to metformin therapy. The authors developed an optimal model demonstrating that a set of four parameters (gender, rs12208357 genotype, familial T2DM back ground, and waist-hip ratio) had the greatest importance for the prediction of the response to metformin therapy in Russian patients.

The study is well designed and executed properly. The manuscript is well written and summarizes the integrated contribution of the genetic factors and phenotypic features to therapeutic response. The study contributes to improving the current understanding of the pathogenesis of type 2 diabetes mellitus and the molecular mechanisms of the response of patients with diabetes to therapy, which may contribute to the development of personalized treatment of the disease. The manuscript may be accepted for publication in its current form.

Comment:

The WHR abbreviation in the Abstract section is not deciphered. Please, correct it.

Reviewer 2 Report

Comments:

1. Page 2, lines 87-88: Authors mentioned: “there are 464 unrelated patients with type 2 diabetes and 129 healthy volunteers were recruited”. But in line 98, they said “299 patients among them took metformin – 131 individuals as monotherapy, and 168 took it in combination with other oral hypoglycemic agents”. How about the rest 165 patients? Do they take metformin, if not, excluding their data from Table 1 is a better choice?

2. Page 3, Table 1: there are big differences between patients and controls, such as age and BMI. How did the authors confirm these characteristics will not contribute to the results?

3. In Table 3, the genotype and allele frequencies of studied variants in non-responder patients should be divided into two groups based on metformin: monotherapy and monotherapy + combination therapy. So, it will be clear the function of metformin except for other drugs.

4. The quality of Figure 1 is low, please adjust the resolution and change a better one.

5. Please explain why they choose 44 cases and 44 controls to construct the training set.

6. Please address the difference between Figure 1a and supplementary S3. What do they mean by “values in 4-fold CV for different types of models (see main text) are shown”?

7. The data shown in Figure 1b reveal that gender is the highest impact on predictive power. Please provide more details on how it works, and provide a table containing the genotype and allele frequencies of studied variants in non-responder patients and glycemic response to metformin treatment based on gender.

8. There are many genetic variants target SLC22A1, such as rs1867351, rs683369, rs3413415, rs2282143 rs34205214, rs34130495, rs34888879, rs72552763, rs35270274, rs41267797 and rs78899680. Please explain why they choose rs628031 and rs12208357?

9. Can authors draw a figure reflecting individual-predicted concentrations for metformin treatment glycemic response vs non-responder patients?

Round 2

Reviewer 2 Report

Agree to accept.

Author Response

We thank you for the careful review of the paper and your constructive remarks